# Automatic Slouching Detection and Correction Utilizing Electrical Muscle Stimulation

Kattoju Ravi Kiran*    Corey.R.Pittman†    Yasmine Moolenar‡    Joseph J.Laviola Jr.§

Interactive Systems and User Experience Lab
University of Central Florida, USA

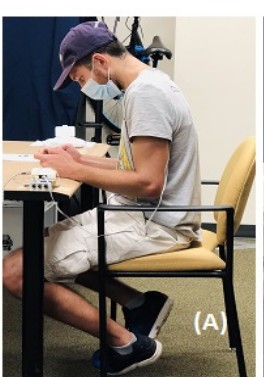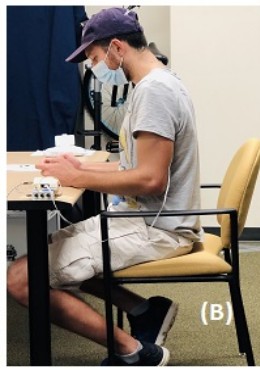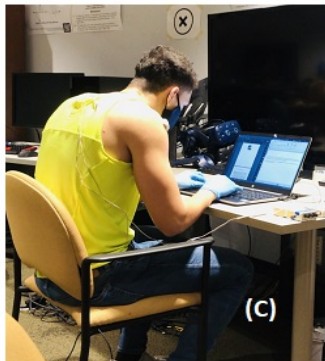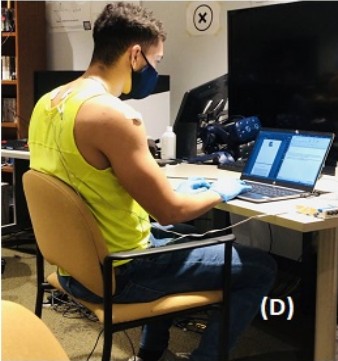

Figure 1: Improper posture can have long term health ramifications. Presented here are images of slouched and corrected posture using Electrical Muscle Stimulation: (A)Mobile Gaming - Slouched posture, (B)Mobile Gaming - Corrected posture, (C)Text Entry - Slouched posture, (D)Text Entry - Corrected posture.

## ABSTRACT

Habitually poor posture can lead to repetitive strain injuries that lower an individual's quality of life and productivity. Slouching over computer screens and smart phones are common examples that leads to soreness, and stiffness in the neck, shoulders, upper and lower back regions. To help cultivate good postural habits, researchers have proposed slouch detection systems which alert users when their posture requires attention. However, such notifications are disruptive and can be easily ignored. We address these issues with a new physiological feedback system that uses inertial measurement unit sensors to detect slouching, and electrical muscle stimulation to automatically correct posture. In a user study involving 36 participants, we compare our automatic approach against two alternative feedback systems and through two unique contexts-text entry and gaming. We find that our approach was perceived to be more accurate, interesting, and outperforms alternative techniques in the gaming but not text entry scenario.

**Index Terms:** Human-centered computing—Human computer interaction (HCI)—Wearable computing—Preventive Healthcare Posture correction—Slouching—Electrical Muscle Stimulation;

## 1 INTRODUCTION

The alignment of body parts, bolstered by correct muscular tension, plays a crucial role in maintaining healthy posture. Poor posture in the workplace leads to Repetitive Strain Injuries (RSI), which in turn leads to health deterioration and low productivity [2]. In addition, long-term musculoskeletal disorders [44] are becoming increasingly prevalent in working populations. Improper occupational standards, poor workstation arrangements, and unnatural gaming positions are often the biggest factors contributing to RSI. Repetitive processes, such as the use of computer systems and smart phones, present a high risk of RSI, examples of which include wrist extension, neck cradling, forward neck, slouching, and uneven weight distribution on the legs. In this paper, we address slouching, which is one of these RSI. The RSI associated with slouching, if not detected, analyzed, and corrected at an early stage, may lead to the development of poor posture habits which induce intense pain, trigger point pain, and muscle tightness in the chest, neck, shoulders, and back regions.

In the United States, nearly $90 billion [8] are spent yearly for the treatment of RSI and lower back pain arising out of poor workplace postures, primarily slouching [10]. Slouching causes muscle dysfunction, or dystrophy of the transverses abdominus muscle [44] responsible for stabilizing the torso in an upright position and is directly associated with lower back pain. Lower back pain / injuries are one of the noted root causes of disability in the world, and affects approximately 80% of world population at some point in their lives [19, 47]. As current intervention technology offers only slouch detection and requires users' conscious effort to correct improper posture, there is a dire need for the implementation of a wearable intervention technology with autonomous capabilities for slouched posture detection and correction which facilitates proper posture maintenance during work and gaming activities. As Electrical muscle stimulation (EMS) can be applied to cause involuntary muscular contractions and generate a physiological response [9, 43, 50], we integrated EMS with a slouch detection system to automatically correct poor habitually slouched posture and restore correct posture through these involuntary muscular contractions. The main contributions of this work include the development of a wearable intervention prototype that autonomously detects and corrects slouched posture through a physiological feedback loop utilizing EMS. We also evaluate the performance of our automatic approach in breaking the habit of slouching, and for training and developing good postural habits.

*e-mail:Kattoju.ravikiran@knights.ucf.edu
†e-mail:cpittman@knights.ucf.edu
‡e-mail:ymmoolenaar@knights.ucf.edu
§e-mail:jjl@cs.ucf.edu

## 2 RELATED WORK

Although, there is a consensus that posture can be improved through ergonomic solutions such as adjusting desk and chair heights, monitor viewing angles, and keyboard and mouse positions [2], only a small number of reliable posture monitoring techniques exist, especially for slouching. Technological advances in wearable technology have attracted increased attention for development of real time posture monitoring, detection and self-correction feedback alert mechanisms [13, 16, 23]. Despite efforts and novelty of wearable intervention technologies, current researchers are focused on developing more accurate monitoring techniques and development of predictive algorithms for better detection of poor posture. As a result, aspects such as user perception, aesthetics, and correction response times are often neglected.

### 2.1 Posture Detection with Real Time Feedback

Most slouching detection techniques employ a variety of sensors such as IMUs, force sensitive resistors, electromagnetic inclinometer [3], fiber-optic sensors [4], cameras [52], and smart garments [1, 12]. Current slouching detection and alert mechanisms employ one of three traditional feedback types via visual, audio, and haptic to convey posture information to the user. Audio feedback was integrated with strain gauges [39], instrumented helmets [56], accelerometers [12], gyroscopes [54], and IMUs [1] for correcting poor posture. Similarly, real time visual feedback utilizing a set of IMUs [16, 23] were also developed for development of good posture. Additionally, both visual and audio feedback systems were integrated with IMU based wearables in *Zishi*, an instrumented vest [53], *Spineangel*, a wearable belt [45], and *SPoMo* [41] and *Limber* [22], smart systems to investigate the relationship between poor posture and lower back pain. Further, hybrid posture detection techniques utilizing electromagnetic technology and accelerometers [3] and inductive sensors [48] were developed to deliver vibrotactile feedback to users for self-correction. Researchers also integrated haptic feedback with IMUs for postural balance [55], gait training [13], and postural control [35]. Finally, a comparative study aimed at improving spinal posture concluded that the IMU-based systems performed more accurately than the vision-based system [52]. Also, a qualitative assessment of different commercially available wearables for posture detection (LUMO Lift, LUMO Back and Prana), and correction through haptic, and visual feedback concluded that haptic feedback enabled a shared responsibility for detecting poor posture and subsequently delivering alerts for self-correction [36].

### 2.2 Electrical Muscle Stimulation

EMS is a non-invasive technique for delivering electrical stimuli to muscles, nerves and joints via surface electrodes positioned on the skin to deliver an acute/chronic therapeutic effect. Traditionally, EMS was utilized for therapeutic pain management to alleviate chronic muscle strain, acute muscle spasms, and in rehabilitation to regain muscle strength [5, 9], and normal movement after injury or surgery [50]. EMS has also been utilized for generating functional movements resembling voluntary muscle contractions like evoking hand opening [9], stimulating reflexes for swallowing disorders [46], enabling neuro-prosthesis control [43], and restoring functions impaired through injury, surgery, or muscle disuse.

### 2.3 EMS in Human Computer Interaction

EMS has found new interest in human computer interaction (HCI) for applications in gaming, augmented reality, and virtual reality (VR) training [20, 24–27, 49]. Integration of EMS with adaptable wearables has enabled development of novel interaction techniques for activity training, immersive feedback technologies and spatial user interfaces. Owing to EMS's ability to invoke involuntary muscular contractions, it has enabled activity training in users for acquiring and developing new skills such as playing musical instruments [51],

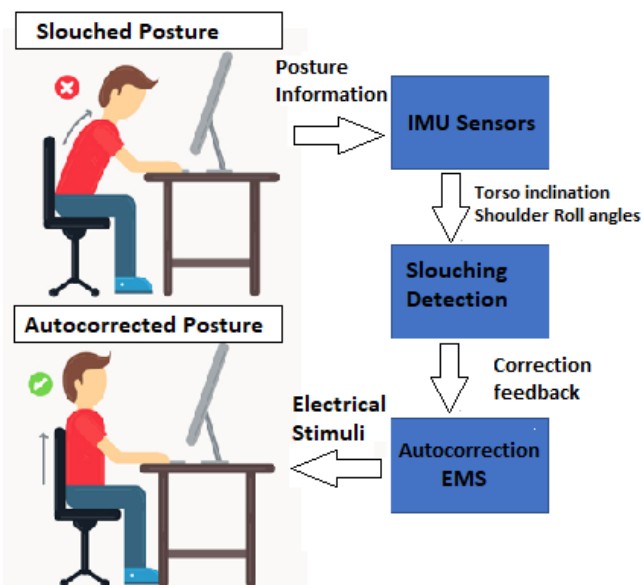

Figure 2: Physiological Feedback Loop: Automatic Slouching Detection and Correction System.

learning object affordance [31], and developing preemptive reflex reactions [18, 37]. Additionally, EMS-based feedback systems have facilitated providing force feedback for impact [11, 28], adding physical forces to mobile devices [34], virtual objects [33], and walls and obstacles [29, 32]. EMS has also enabled delivering more immersive virtual experiences for sharing and augmenting kinesthetic feedback from hand tremors in Parkinson's disease [38], actuating emotions of fear and pain in *In-pulse* [21], and transferring emotional states between users in *Emotion Actuator* [15]. Further, adaptability of EMS technology has permitted development of a physiological feedback loop and enabling its integration with input and output interfaces in proprioceptive interaction (*Pose-IO*) [30], actuated navigation [42], actuated sketching [34], unconscious motor learning [6], co-creating visual art works [7], and delivery of discrete notifications [14]. These interactive applications and adaptability of EMS based technology demonstrate its capability in delivering implicit, discrete, and more defined feedback compared to audio, visual and haptic feedback mechanisms while exhibiting portability and hardware miniaturization compared to actuated motors in exoskeletons. The current literature indicates that posture correction systems based on traditional feedback techniques relied completely on the users' willingness to correct improper posture and that posture correction utilizing EMS has not been fully explored. This presents an opportunity for developing novel wearable intervention technology for autonomous detection and correction of slouched posture.

## 3 AUTOMATIC SLOUCHING DETECTION AND CORRECTION UTILIZING EMS

We developed a physiological feedback loop-based wearable intervention prototype relying on IMU sensors and EMS (illustrated in Figure 2). Our prototype employed three Metawear MMC wireless sensors for measuring angular changes, and the openEMSstim package [26] for presenting the EMS feedback. We developed a user interface using the Metawear C# SDK and integrated the EMS hardware to complete the physiological feedback loop. As slouching was mainly characterized by torso inclination and forward rolling of the shoulders [17], IMU 1 was placed at the center of the collar bone above the chest (illustrated in Figure 3 (B)) and the other IMU's 2 and 3 were placed on the center of each deltoid (illustrated in Figure 3 (A) & (C)).

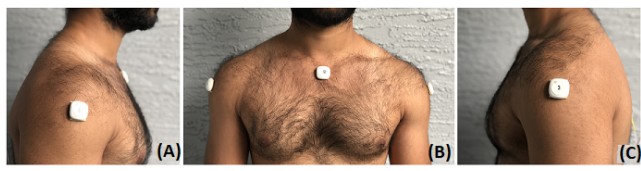

Figure 3: Wireless IMU sensor placement for posture monitoring and detecting slouched posture: (A)Side view showing sensor placement on left deltoid, (B)Front view showing sensor placement below center of collar bone above the chest, (C)Side view showing sensor placement on right deltoid.

Figure 4: EMS electrode placement on rhomboid muscles for autocorrection using EMS feedback.

The change in posture was calculated from the angular information obtained from the IMU sensors. The user's torso inclination angle was calculated from the pitch of IMU1, and the roll angles on the shoulders were calculated from the roll of IMU's 2 and 3. Our system detected slouching when the user's current torso inclination and shoulder roll angles, both approached and remained at a threshold level for a period of 5 seconds. The threshold level is preset as $-3°$ of the torso inclination and shoulder roll angles recorded in the slouched position during calibration. $-3°$ was chosen to overcome measurement errors without increasing false positives and a 5 second time duration ensured random movements do not lead to false positive slouch detection. These design choices were validated during our pre-study trials. The threshold angle of $-3°$ was used to initiate the 5 seconds timer, and the slouch angles detected were recorded at the end of the timer when the feedback was presented. The purpose of the timer was to ensure false positives due to participant behavior do not trigger the feedback response.

When slouching was detected, the automatic correction feedback was presented by applying electrical stimulus to the rhomboid muscles (illustrated in Figure 4) for generating a pulling force in the opposite direction from the slouched posture and thereby, generating a physiological response to bring the user back to the upright or correct position. Two pairs of electrodes were utilized for contraction of the rhomboid muscles which causes the shoulders blades to be pulled back, thus unrolling the shoulders and bringing the torso back to the upright posture. IMU and EMS calibration play a crucial role in the effectiveness of the system. The calibration process included correcting IMUs offset value in the upright position of the user and recording the angular change in the slouched posture with respect to the upright position. The EMS intensity calibration was manually incremented to deliver an intensity that was optimal for generating involuntary muscular contraction and avoid any pain. This EMS intensity provided to the user for generating the necessary pulling force for correcting the slouched posture and restoring the upright position was recorded and utilized during the experiment. The TENS device was able to deliver intensities between (0-100mA). A continuous 75 Hz square wave pulse at the recorded EMS intensity and a pulse width of 100 $\mu$s was supplied as the electrical stimulus to the users.

## 4 METHODS

The goal of our study was to evaluate the overall effectiveness and user perception of an automatic slouch detection and correction feedback system (EMS) compared to traditional audio and visual

Table 1: User ranking on posture awareness, devices and EMS.

| User Experience | Application | Mean | S.D |
|---|---|---|---|
| Exposure to Posture Alert Devices | Text Entry | 1.61 | 1.16 |
| | Mobile game | 1.89 | 1.09 |
| Exposure to EMS | Text Entry | 2.33 | 1.37 |
| | Mobile game | 2.06 | 1.43 |
| Experienced posture problems | Text Entry | 4.22 | 1.55 |
| | Mobile game | 4.33 | 1.49 |
| Experienced slouching | Text Entry | 5.06 | 1.50 |
| | Mobile game | 5.4 | 0.96 |

*Note: User experience ranking based on a 7-point scale where 1 means never / no experience and 7 means frequently / very experienced.*

feedback modalities requiring self-correction by the user based on audio and visual notifications, respectively. We also identified two common causes of slouching in day to day activities such as computer related workplace tasks, and mobile gaming [17, 40], and investigated our automatic approach using EMS across these common causes of slouching. Our objective was to determine if our automatic posture detection and correction system using EMS would be a viable technique for correcting slouched posture as compared to the visual and the audio feedback channels while being engaged in a computer related workplace task and playing a mobile game.

### 4.1 Subjects and Apparatus

We recruited 36 Participants (Male=31, Female=5) for the study with 18 participants for each application- text entry and mobile game. All participants recruited were above the age of 18 years and the mean age of participants was 22.05 years ($S.D. = 3.13$). All participants were able bodied and had corrective 20/20 vision. We used three Metawear MMC IMU sensors for monitoring the torso inclination angles and the shoulder roll angles. The EMS was generated with an off-the-shelf Tens Unit (TNS SM2MF) and controlled by the openEMSstim package for activating and modulating the intensity of the electrical stimuli supplied to the muscles. The hardware used for the text entry application was a 14" Intel i7 Laptop, and a 2nd generation iPhone SE was used for the mobile game application. From the pre-questionnaires, participants ranking of their prior exposure to posture alert devices and EMS, and experience with posture problems and slouching are noted and illustrated in Table 1. Participants ranked their exposure and experience on a 7-point scale with 1 meaning never/no experience and 7 meaning frequently/very experienced.

### 4.2 Experimental Design

A 2 by 3 mixed subjects experiment with 36 participants was conducted to investigate the performance and feasibility of our approach. The within subject factor was feedback type (visual, audio, and EMS) and the between subject factor was application type (text entry, and mobile game tasks). We compared the performance of automatic slouching correction using the EMS feedback against the self- correction in the visual and audio feedback techniques. Average correction response times and user perception of the system across the two applications and three feedback types were also evaluated. In the text entry application, users were required to complete a text entry task and the mobile game application required the users to play a mobile based Battle Royale game called "PlayerUnknown's Battlegrounds (PUBG) Mobile[1]". PUBG was selected based on its popularity (400 million players), level of engagement and demographics (people aged between 15-35 years who may be prone to long working and gaming sessions). In both applications, the users were required to complete all three modalities:

---

[1]https://www.pubg.com/

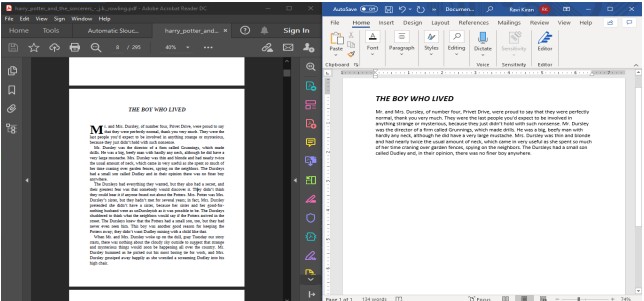

Figure 5: Text entry study showing 50-50 split screen with a PDF document (zoom set to 40%) on the left and a Microsoft Word document (zoom set to page width) on the right. Participants were required to read from the PDF document and type in to the Word document.

- Modality 1: Visual alert feedback and self-correction
- Modality 2: Audio alert feedback and self-correction
- Modality 3: EMS feedback and automatic correction

In each application, participants were required to complete all three modalities in a counterbalanced order to minimize learning effects. The independent variables in the study were the three different modalities and the dependent variables were the average correction response times, and user perception parameters such as overall experience, accuracy of correction feedback, engagement and task disruption, and comfort. Each study session lasted approximately 75 minutes and the participants were compensated $15 for their participation. This study was approved by the Institutional Review Board of the University of Central Florida.

### 4.3 Research Hypotheses

Our study was designed to determine the effects of automatic or self-posture correction on user experience across the two applications, and three modalities. As such, we expect significant differences across the three modalities which could influence user experience. For investigating into the user perception, we have five research hypotheses with two parts namely, (a) in text entry, and (b) in mobile game.

- **H1:** In the text entry (a) or the mobile game (b), the average correction response time to slouching feedback will be faster in EMS feedback compared to the visual and audio alert feedback.
- **H2:** In the text entry (a) or the mobile game (b), the user perception of accuracy of slouching posture correction in EMS feedback will be greater than visual and audio alert feedback.
- **H3:** In the text entry (a) or the mobile game (b), comfort in EMS feedback will not be significantly different compared to visual and audio alert feedback.
- **H4:** In the text entry (a) or the mobile game (b), no evidence will be found for a difference in task disruption across the visual, audio, and EMS feedback
- **H5:** In the text entry (a) or the mobile game (b), automatic correction using EMS feedback delivers better user experience compared to visual and audio alert feedback.

### 4.4 COVID-19 Considerations

Due to the ongoing COVID-19 pandemic, we wanted to ensure safety for the participants and researchers. Following our institutions guidelines, all individuals were required to always wear face masks. Between each participant, we sanitized all devices and surfaces that the participants and researchers would be in contact with, to ensure safety during the study. Furthermore, all users were required to wear a face mask to participate in the study. We also provided hand sanitizer, cleaning wipes, and latex gloves to reduce risk of contracting the disease. Though we cleaned all surfaces between participants, we allowed participants to clean devices as desired.

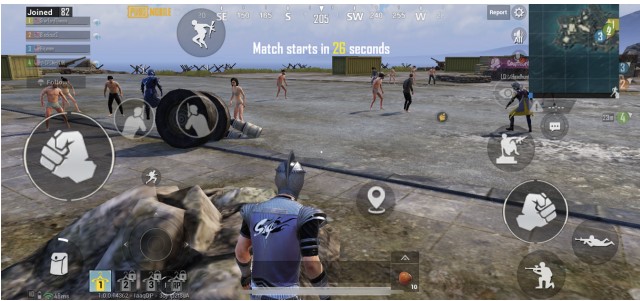

Figure 6: Mobile game study showing lobby area of PUBG mobile prior to start of the game.

### 4.5 Experimental Procedures

Prior to starting the experiment, participants reviewed the consent form that details the experiment, safety, risks, compensation, and compliance, and were required to provided verbal consent for the study session to start. Participants then completed a survey on their knowledge and experience on workplace related posture issues, intervention technology and EMS. Next, IMU sensors were placed on the participants on their deltoids and center of the collar bone above the chest (as shown in Figure 3) for detecting slouched posture and data collection. Adhesive EMS electrodes were placed on the rhomboid muscles prior to the EMS feedback session for correcting slouching (as shown in Figure 4). Subsequently, IMU sensors were corrected for offset, and calibrated with participants seated in upright and slouched positions and with their hands placed on the keyboard or holding the smartphone. During calibration, participants emulated slouched positions by inclining their torso and rolling their shoulders forward. These upright and slouched posture angles were recorded.

Before the EMS feedback session, an EMS intensity calibration process was done manually for each participant, and moderators incremented the intensity until an involuntary muscular contraction causing posture correction is affected. The participants were calibrated manually only once for EMS intensity to generate a physiological response of sitting upright. During calibration, participants were asked to slouch, and moderators manually incremented the EMS intensity. As EMS also produces a tactile or haptic effect even at low intensities, participants were asked to not respond to the tactile or haptic effect to ensure the haptic/tactile component of EMS does not contribute to the automatic correction process in any way. Moderators additionally asked participants to verbally respond specifically to the following questions during calibration to ensure rhomboid muscular contraction and participant comfort: 1) when they initially felt the stimulation (haptic sensation), 2) when the intensity was generating an involuntary muscular contraction and/or when they are experiencing the pulling force towards the upright posture, 3) when any pain is experienced. For each participant, when involuntary muscular contraction was confirmed verbally by the participant and visually verified by the moderators, the optimal EMS intensity that was generating an involuntary muscular contraction to correct the slouched posture, was recorded, and selected for the EMS part of the study.

The above steps are similar for both the text entry and the mobile gaming applications. In the text entry task, participants were asked to read from a PDF document and type into a word document. The PDF and word documents were presented in a 50-50 split screen. For the purpose of conducting the study, the PDF zoom was set to 40% to promote or cause slouching while reading (illustrated in Figure 5). In the mobile game task, participants were asked to play PUBG mobile (illustrated in Figure 6). In both applications, the user's posture was monitored for slouching. The study comprised of three parts: visual, audio, and EMS feedback. Each part of the study is 15 minutes in duration and all participants were required to

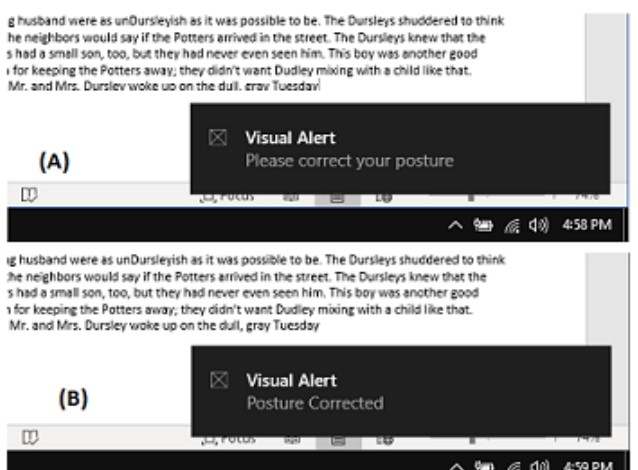

Figure 7: Text entry study-visual feedback: showing Windows 10 pop-up visual notification on the bottom right of the screen. (A) To correct posture when slouching is detected. (B) After posture has been corrected.

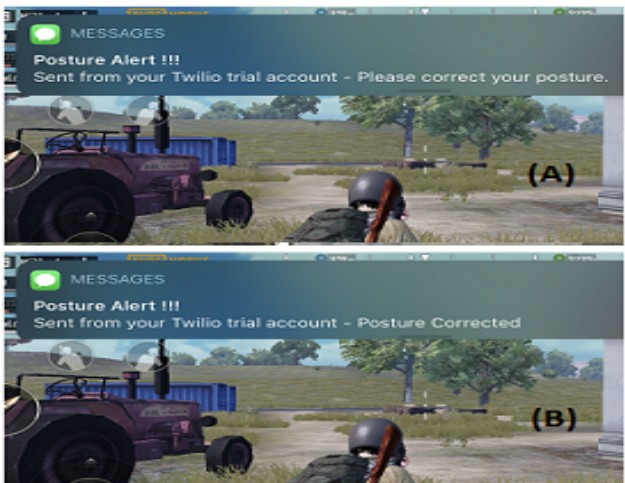

Figure 8: Mobile game study-visual feedback: showing visual notification badges drop down from the top of the display. (A) To correct posture when slouching is detected. (B) After posture has been corrected.

finish all three parts to complete the study. Participants completed a survey about their experience after each part and a comparative survey on their overall experience at the end of the study. All data from sensors and EMS were recorded for analysis and reporting.

### 4.5.1 Visual feedback and self-correction

*Text Entry Application:* When slouching was detected by the system based on the IMU sensor feedback, a Windows 10 visual popup notification "Please correct your posture" is displayed on the bottom right corner of the monitor (illustrated in Figure 7a) and the users were required to sit upright and self-correct their slouched posture till a second visual popup notification "Posture corrected" is displayed to the user (illustrated in Figure 7b). The response times for correcting the slouched posture were recorded.

*Mobile Game Application:* When slouching was detected by the system based on the IMU sensor feedback, an SMS is sent from the C# application to the smart phone with the message "Posture alert: Please correct your posture" and is displayed as a drop down badge notification on the smartphone (illustrated in Figure 8a). After receiving the visual alert notification, the users were required to sit upright, and self-correct their slouched posture till another SMS containing the message "Posture corrected" is displayed to the user (illustrated in Figure 8b). The response times for correcting the slouched posture were recorded.

### 4.5.2 Audio feedback and self-correction

*Text Entry Application:* When slouching was detected by the system, an audio notification "Please correct your posture" is activated and the users were required to sit upright, and self-correct their slouched posture till a another audio notification "Posture corrected" is presented to the user. The response times for correcting the slouched posture were recorded.

*Mobile Game Application:* When slouching was detected by the system, an audio notification bell sound is activated and the users were required to sit upright, and self-correct their slouched posture till another audio notification bell is activated for the user. The response times for correcting the slouched posture were recorded.

### 4.5.3 EMS feedback and auto-correction

*Text Entry and Mobile Game Applications:* When slouching was detected by the system, the EMS is activated to apply the recorded EMS intensity to the rhomboid muscles to invoke an involuntary muscle contraction. This muscle contraction produces a pulling force in the opposite direction to the slouched posture and to generate the physiological response of sitting upright by correcting the torso inclination and shoulder roll caused by slouching. Figure 1(A) and (C) illustrate the slouched posture during the mobile game and the text entry studies respectively. Figure 1(B) and (D) illustrate the corrected posture after EMS has been applied in the mobile game and the text entry studies respectively. The EMS is deactivated once the upright position is detected. The response times for correcting the slouched posture were recorded.

## 5 RESULTS

The average number of slouches in the text entry condition was (7.72, 10, and 8.72) for the audio, visual and EMS feedback modalities respectively and (7.05, 9.11, and 8.38) for the audio, visual and EMS feedback modalities, respectively in the mobile game condition. The average torso inclination and shoulder roll angles were recorded for slouched posture during the calibration process and utilized for detection of slouching which are illustrated in Table 2. For text entry, mean torso inclination angle was $21°$ ($S.D = 3.88°$), while the mean shoulder roll angle was $15.1°$ ($S.D = 3°$). For the mobile game, mean torso inclination angle was $18.24°$ ($S.D = 2.8°$), while the mean shoulder roll angle was $13.84°$ ($S.D = 2.22°$). For the text entry application, the mean electrical stimulation intensity required to correct slouched posture was $39.72mA$ ($S.D = 13.17mA$) while for the mobile game task, the mean electrical stimulation was $47.22mA$ ($S.D = 11.08mA$).

Table 2: Average slouch angles in degrees.

| Slouch angle | Application | Mean | S.D |
|---|---|---|---|
| Torso Inclination angle | Text Entry | 21.00° | 3.88° |
| | Mobile game | 18.24° | 2.80° |
| Shoulder roll angle | Text Entry | 15.10° | 3.00° |
| | Mobile game | 13.84° | 2.22° |

To address H1, a one-way repeated measures ANOVA was performed on the influence of correction feedback type on the average correction response times taken for correcting detected slouched postures after correction feedback is presented to the user in the text entry and the mobile game tasks separately. To address H2 through

Table 3: Friedman test results on the user ranking for H2–H5.

| User perception | Application | $\chi^2$ | $p$ |
|---|---|---|---|
| Accuracy of Correction Feedback | Text Entry | 3.592 | 0.166 |
| | Mobile game | 7.259 | 0.027* |
| Comfort | Text Entry | 1.345 | 0.510 |
| | Mobile game | 4.550 | 0.103 |
| Task Disruption | Text Entry | 0.092 | 0.955 |
| | Mobile game | 5.607 | 0.061 |
| Experienced slouching | Text Entry | 0.407 | 0.816 |
| | Mobile game | 0.400 | 0.819 |

*Note: ∗ indicates significant difference $P < 0.05$.*

H5, non-parametric Friedman tests of differences among repeated measures was conducted on the users' ranking of effectiveness of correction feedback, comfort, task disruption and overall experience. Wilcoxon signed rank tests were performed if significant differences were found. The results were consolidated and presented in Table 3. For H1(a), all effects were statistically significant at the .05 significance level. The main effect for the correction feedback type yielded $F(2,34) = 5.382, p < .05$, indicating a significant difference between visual feedback ($M = 3.86, S.D = 1.27$), audio feedback ($M = 3.9, S.D = 1.28$) and EMS feedback ($M = 2.89, S.D = 1.74$). The average correction response times were faster for EMS feedback than the visual feedback ($t_{17} = -0.961, p < 0.05$), but no significant differences were found between EMS and audio feedback types, and between visual and audio feedback types. For H1(b), all effects were statistically significant at the .05 significance level. The main effect for the correction feedback type yielded $F(2,34) = 20.66, p < .001$, indicating a significant difference between visual feedback ($M = 5.98, S.D = 2.4$), audio feedback ($M = 4.44, S.D = 0.75$) and EMS feedback ($M = 2.70, S.D = 1.04$). The average correction response times were faster for audio feedback than the visual feedback ($t_{17} = -1.538, p < 0.05$), the EMS feedback was faster than Visual feedback ($t_{17} = -3.276, p < 0.01$), and also faster than the audio feedback ($t_{17} = -1.737, p < 0.001$). The post-hoc analysis between the three feedback types shows that the hypothesis H1(a) tested false in that the average correction response times were faster in the EMS feedback type compared to the visual feedback but not the audio feedback. In the case of H1(b), the hypothesis tested true, in that the average correction response times were faster in the EMS feedback type compared to the visual, and audio feedback types as illustrated in Figure 9 (A) and (B).

For H2(a) and (b), the test rendered $\chi^2 = 3.591, p = 0.166$ which was insignificant ($p > .05$) for the text entry application, while for the mobile game application, the test rendered ($\chi^2 = 7.259, p = 0.027$) which was significant ($p < .05$). A post-hoc analysis with Wilcoxon signed-rank tests was conducted for the mobile game application with a Bonferroni correction applied, resulting in a significance level set at $p < 0.017$. Median perceived accuracy of slouching correction for the Visual, Audio, EMS feedback were $6, 6, 7$ respectively. There was a statistically significant difference between the visual and the EMS correction feedback type ($Z = -2.591, p = 0.010$), and also between EMS and audio correction feedback type ($Z = -2.585, p = 0.010$). However, there was no significant difference between audio and visual correction feedback types ($Z = -0.942, p = 0.346$). Therefore, H2(a) tested false and indicated that the users perceived all three feedback types equally accurate in the text entry application. Whereas H2(b) tested true and indicated that the users perceived that the accuracy of EMS correction feedback was more effective than the visual and the audio feedback in the mobile game application. For H3(a) and (b), the test rendered $\chi^2 = 1.345$ and $p = 0.510$ which was insignificant ($p > .05$) for the text entry application, while for the mobile game application, the test rendered $\chi^2 = 4.550$ and $p = 0.103$ which was

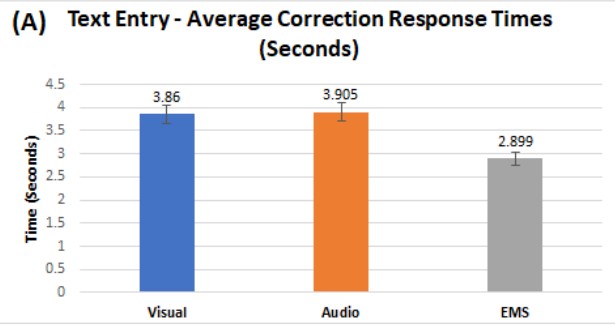

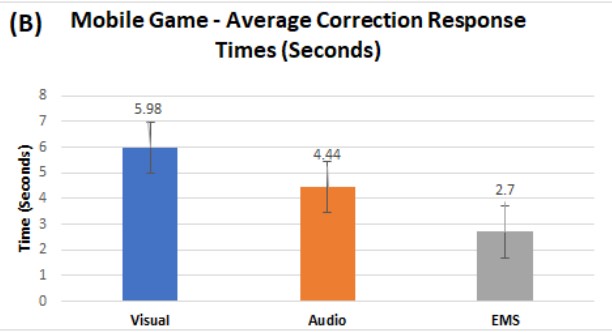

Figure 9: Average Correction Response Times (in Seconds) across (A) Text Entry and (B) Mobile Game for all correction feedback types - (1) Visual, (2) Audio, (3) EMS. Error Bars: 95% CI.

insignificant ($p > .05$). Therefore, both H3(a) and (b) tested true and indicated that users perceived all three feedback types equally comfortable in the text entry and the mobile game application. For H4(a) and (b), the test rendered $\chi^2 = 0.092$ and $p = 0.955$ which was insignificant ($p > .05$) for the text entry application, while for the mobile game application, the test rendered $\chi^2 = 5.607$ and $p = 0.061$ which was insignificant ($p > .05$). Therefore, both H4(a) and (b) tested true and indicated that users perceived EMS correction feedback's disruption no worse than the other two feedback types in the text entry and the mobile game application. For H5(a) and (b), the test rendered $\chi^2 = 0.407$ and $p = 0.816$ which was insignificant ($p > .05$) for the text entry application, while for the mobile game application, the test rendered $\chi^2 = 0.400$ and $p = 0.819$ which was insignificant ($p > .05$). Therefore, both H5(a) and (b) tested false and indicated that users perceived overall experience across all feedback types equally well in the text entry and the mobile game application.

Participants ranked their shared responsibility with auto-correction utilizing EMS on a 7-point scale where 1 means not at all and 7 means completely. The mean shared responsibility exhibited by the users was 2.77 ($S.D = 1.7$) in the text entry task, while for mobile game condition, users reported that they helped/ aided auto-correction with a mean shared responsibility of 2.5 ($S.D = 0.95$). Participants also ranked how interesting the EMS concept was to use for posture correction on a 7-point scale where 1 means not at all and 7 means completely. The mean ranking received for EMS concept being interesting was 6.58 ($S.D = 1.01$). 27 out of 36 users reported that they would purchase EMS feedback for slouched posture correction if it were commercially available. Participants' responses when asked to comment on their experience with EMS showed that EMS feedback felt *"more natural"*, *"not easily ignorable"* and better than audio and visual modalities as they cause *"over/ under correction"* of posture. Additionally, participants also reported about the EMS feedback that *"the system accurately initiated the stimulus when slouched and stopped after posture was corrected."* and that EMS *"would enable me to not worry about my posture during highly en-*

*gaging tasks."* One user responded that EMS is *"unobtrusive and discrete method of auto-correcting posture."* and EMS was the *"least disruptive"*. Further, users reported *"I cannot listen to some one while i am trying to read and type.",* and *"the visual notifications were annoying and distracting when i was typing."* indicating that the audio and visual feedback were placing a cognitive load on the user. Other user comments include *"this can be a good training device but EMS requires getting used to"*, *"it actively and immediately corrected my slouched posture"*, *"training device for maintaining proper posture"*, *"the tingling sensation feels weird but good"*, *"this can seriously help people with posture problems."*

## 6 DISCUSSION

While slouching detection and alert systems were designed and tested, we note that posture correction response times and user perception of the systems have not been measured or reported. Therefore, our study focused on evaluating the performance, and user perception of our autonomous system for detecting and correcting slouching. Our automatic slouching correction using the EMS feedback system outperformed the visual and audio feedback types based on the average correction response times suggesting that visual and audio feedback place an additional cognitive load on the user while being engaged in their task and rely completely on the user's willingness to self-correct their posture. However, as EMS feedback does not require the user's attention, this has made it significantly faster than the other two feedback types.

Users also perceived that EMS feedback corrected their posture more accurately than the other two feedback types that required self-correction. Users reported that self-correction in the visual and the audio feedback types caused them to always over-correct their posture as their awareness of it was minimal while being engaged in the task at hand. Whereas EMS feedback did not require the user's attention and always accurately activated when the slouched posture was detected and deactivated after a posture had been corrected. The user rankings on accuracy of EMS feedback indicated that EMS feedback was perceived to be more accurate in the mobile game application than the text entry application. This interesting finding may have been due to different factors such as nature of the two applications, complexity of the task, users' connection to the device and varied range of motion involved in auto-correction using EMS feedback across the two applications. It was also interesting to note that EMS feedback and auto-correction were perceived equally comfortable and no more disruptive than traditional visual and audio alert feedback but with the added advantage of automatic correction. This may have been because EMS feedback relied entirely on the user's physiology and careful EMS intensity calibration with user feedback on their level of comfort to deliver a somatosensory feedback that discretely enabled posture awareness without disruption. With regards to comfort, users reported that their awareness of the IMUs and EMS electrodes on their body was minimal suggesting that our prototype could be a viable wearable intervention device for posture correction.

Further, shared responsibility in aiding the auto-correction using EMS was exhibited and reported by the users suggesting that the sensory confirmation delivered by activation and deactivation of EMS encouraged their involvement in the posture correction process and increased their posture awareness. This demonstrated that users can adapt to the system and gradually utilize it as a training device for development of good postural habits in the long run. The EMS intensity required for auto-correction in the mobile game task was higher than the text entry task, suggesting that the task type, level of engagement, range of motions involved in the correction process influence the EMS intensity required for correcting different levels of slouching. As shown in the results section, the EMS intensity varied across the study population. This could be due to factors such as different body types, muscle physiology, and activity levels.

Finally, users perceived that EMS was an interesting concept to use for automatic posture correction while they were engaged in their tasks. 20 out of 36 users reported that EMS feedback and auto-correction was their most preferred feedback type while 75% of the study population was willing to purchase the automatic correction using EMS feedback if it were a commercially available product. Therefore, our autonomous system could be a valuable alternative or an addition to existing environment, health, and safety (EHS) protocols at workplaces for enhancing productivity, worker health and in preventive health care.

## 7 LIMITATIONS AND FUTURE WORK

Although limitations include placement of the sensors and electrodes, we plan to integrate them into wearable clothing and devising an auto-calibration system that can be customized to each individual's comfort. Another limitation of the current study was the gender of the users, the proportion of the male users was higher than the female users (Male=31, Female=5). We plan to conduct a further study to balance the male to female ratio. Our future work includes testing the automatic detection and correction of slouching under different physical conditions such as standing, walking, and carrying different loads on users' backs.

## 8 CONCLUSION

We have demonstrated that our physiological feedback loop based on automatic slouching detection and correction with EMS is a viable approach to supporting posture correction. Our auto-correction system utilizing EMS feedback demonstrated significantly faster posture correction response times compared to the self-correction in the visual and audio feedback. Our approach also showed that users perceived EMS feedback to be more accurate, just as comfortable and produced no more disruption than the alternative techniques it was tested against in both the text entry and the mobile game applications. Therefore, automatic slouching detection and correction utilizing EMS shows promising results and can be developed as an alternative method for posture correction. Our approach could prove useful in preventive healthcare to avoid workplace related RSI and be particularly beneficial to people involved in highly engaging tasks such as gaming, diagnostic monitoring, and defense control tasks.

### ACKNOWLEDGMENTS

This work is supported in part by Army RDECOM Award W911QX13C0052 and NSF Award IIS-1917728. We also thank the anonymous reviewers for their insightful feedback.

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
