# OpenReview forum: "Automatic Slouching Detection and Correction Utilizing Electrical Muscle Stimulation"
_graphicsinterface.org/Graphics_Interface/2021/Conference — GI 2021_

### Official Review · AnonReviewer2 · 2021-01-13
**First steps towards an in-depth investigation of automatic posture correction through ems**

**Rating:** 5
**Confidence:** 4

**Review:**

This paper reports on an early prototype and user evaluation of a wearable system that exploits electrical muscle stimulation to automatically correct slouched posture. The exploration of physiological feedback for the design of new interactive systems is a compelling research area and indeed research is needed to understand the integration of interactive technology with the human body. As such, the research is original in that it investigates the use of ems as a mean for automatic posture correction: previous studies focused on sensors systems for posture monitoring, with audio and visual and haptic feedback to promote voluntary posture correction.

The work, however interesting, is still in its infancy. With the evaluation, authors show that EMS is promising, but unfortunately, the most critical issues are only mentioned in the future work section and the discussion of the results is shallow. The length of the paper, therefore, does not match with the contribution: more work is needed to strengthen (1) the design and technical implementation of the system, as well as (2) the experimental evaluation and (3) the implications of the results.

1.	The implementation of the system seems very straightforward with respect to the design choices to detect a slouch posture. The system look very much like an early prototype that can be used in early design stage to gather requirements or test design ideas. I was expecting many of the functionality reported as future work to be implemented in the system.
The threshold level and the time period to model the slouch posture seem to have been determined empirically, that the authors report to validate in a pre-study trial. I miss a detailed account on the empirical testing and the choices the authors made to finally opt for the selected sensor setup.
I also wonder why the authors used SMS as a mean to trigger visual information on the mobile device. SMS delivery introduces an unreliable delay that affects the response time. How did the authors take that into account?

2.	There are already IMU-based systems for slouch detection and the contribution of the paper relies on the use of EMS as an autocorrection mechanisms. In this respect, authors use visual and audio feedback as baseline for their evaluation. I wonder why they didn’t compare their system with haptic feedback solutions.
Authors report that “As EMS also produces a tactile or haptic effect even at low intensities, participants were asked to not respond to the tactile or haptic effect to ensure haptic/tactile component of EMS does not contribute to the automatic correction process in any way”. How can the authors be sure that they isolated the user response to haptic feedback? The authors report in the text that during the experimentation users seemed to learn how to interact with the system, which suggests that participants responded to the haptic feedback of the ems and create a sort of symbiotic interaction: a conscious autocorrection. This is a very interesting area to investigate and more longitudinal studies are needed to understand to what extent autocorrection mechanisms and haptic feedback (maybe together with audio and visual) can be designed to raise users awareness toward to their posture. Increasing awareness is only briefly mentioned in the discussion.
I also have issues with the experimental setup. What is the effect of the hardware setup (laptop vs mobile) on users’ posture? While I agree that the task are representative, the physical setup is not in the case of the mobile device. Were the participants able to move at least the device freely (e.g., bring the device closer to their head)? From Figure 1 it seems that participants had to interact with the device with their hands touching the table.

3.	I miss, in the discussion, the authors’ interpretation of the difference in the intensity of the electrical stimulation for the two conditions. Was it something that depended on the physical setup of the experimental tasks?
I also miss a more in depth discussion that resulted in insightful findings on the user experience that ultimately would inform the design of the wearable system. For example, the authors report that participants perceived ems feedback as more accurate. Neglecting the fact that there is no definition of what “accurate” means in this context, the authors only list some factors that “may have” cause it, which makes the finding not particularly meaningful.
Minor issues:
I suggest authors to reduce the verbosity of the paper. There is room to reduce the state of the art and also there are repetitions in the text that could be removed, thus helping to shorten the paper and improve the clarity. For example the sentence “Seventy- five percent of the study population (27 out of 36 users) reported that they would purchase EMS feedback for slouched posture correction if it were commercially available“ appears twice in the text.

---

### Official Review · AnonReviewer3 · 2021-01-13
**Novel technique and good study**

**Rating:** 8
**Confidence:** 4

**Review:**

The paper "Automatic Slouching Detection and Correction Utilizing Electrical Muscle Stimulation" presents a technique for detecting and correcting a slouching posture when seated at a desk. The technique uses a combination of IMU sensors and electrical muscle stimulation (EMS). A study is presented where EMS is compared to visual feedback when slouching is detected. Two tasks are studied, one where the participants are doing word processing on a laptop and one where the participants are playing a mobile game. Results show that slouching is corrected more quickly and accurately using EMS and that the participants preferred this approach over visual feedback.

The paper is very well written, well structured, and easy to follow. The technique presented is well-designed and the comparison between visual feedback and EMS is convincing. Generally, the explanations in the paper are very pedagogical. The coverage of related work is good, and it is clear to see that there's a novel contribution in the paper.

One thing I was missing from the paper was an understanding of how prevalent slouching is. It could, e.g., have been interesting to have a baseline in the study where only slouching detection was monitored without any feedback to correct it. This way, it would be possible to assess how much the technique improved overall posture.

Overall, I recommend accepting this paper.

---

### Official Review · AnonReviewer1 · 2021-01-13
**EMS-based posture correction technique**

**Rating:** 7
**Confidence:** 3

**Review:**

This paper presents a nice interface design, implementation, and study relating to using Electrical Muscle Stimulation - where electrical pulses create involuntary muscle contraction - to support healthy posture awareness and correction in adults. I liked reading this paper - it was a simple idea, a nice thorough background treatment, and detailed implementation and study sections. I recommend it for publication.

Although my initial reaction was to envision people being shocked and forced into uncomfortable positions (laugh), the fact that the sensation is a simple pulling and the user still has to fully correct their posture, is key to this actually being viable.

I think the biggest weakness of the work is perhaps the verbose writing. It is quite long winded, and without removing any actual content I think the paper could be shortened significantly, increasing the potential for broader reading and thus its impact.

I do have some comments
 - Please clarify if this work received institutional ethics clearance. It is hinted at in the paper but I couldn't find it mentioned explicitly.
 - I'm not sure the up-front framing does the work justice. The abstract highlights user perceptions, but the paper also found actual results on response time and posture accuracy. Perceptions, of course, are highly biased given that participants likely know your favorite. I would suggest couching analysis / presentation of user perceptions in a relevant way, e.g., may relate to willingness to adopt.
 - the RW is a bit too list-like. While this is helpful as a resource, I felt it would be stronger with a more thematic approach - may also help condense the writing.
 - For H4, your hypothesis is that "will not be significantly different". I think this test actually requires equivalence testing, which is not what you did. You may want to revise the hypothesis to match what you did, e.g., "no evidence will be found for a difference between...". Subtle but important difference, I think.
 - for the post-hocs did you do family-wise correction?
 - I was surprised to see little negative discussion from the participants on the EMS system. really?
 -

---

### Meta-Review · Area_Chair1 · 2021-01-14

**Recommendation:** Accept
**Confidence:** 5

**Metareview:**

The three reviewer agree on the novelty of the proposed technique as well as on its relevance.
The paper clearly falls in the acceptance side.
R1 and R2 are more critical: among reviewers' suggestions to improve the paper, authors should definitely consider to:
 - Significantly shorten the paper: it is too verbose (R1) and the length of the paper doesn't match the contribution (R2). I would say that the paper can be easily shorten to 5/6 pages.
-  Correct H4 as reported by R1
-  Clarify if the work received ethics clearance (R1)
-  Deepen the discussion on the user experience (R1 and R2)

---

### Decision · Program_Chairs · 2021-01-16

Accept